# Deep BiLSTM Attention Model for Spatial and Temporal Anomaly Detection in Video Surveillance

**DOI:** 10.3390/s25010251

**Published:** 2025-01-04

**Authors:** Sarfaraz Natha, Fareed Ahmed, Mohammad Siraj, Mehwish Lagari, Majid Altamimi, Asghar Ali Chandio

**Affiliations:** 1Department of Information Technology, Quaid e Awam University, Nawabshah 67450, Pakistan; fajokhio@quest.edu.pk (F.A.); legharimehwish@quest.edu.pk (M.L.); asghar.ali@quest.edu.pk (A.A.C.); 2Department of Software Engineering, Sir Syed University of Engineering & Technology, Karachi 75300, Pakistan; 3Department of Electrical Engineering, College of Engineering, King Saud University, Riyadh 11543, Saudi Arabia; mtamimi@ksu.edu.sa

**Keywords:** convolutional neural network, recurrent neural network, BiLSTM, multi-attention layer, anomaly detection

## Abstract

Detection of anomalies in video surveillance plays a key role in ensuring the safety and security of public spaces. The number of surveillance cameras is growing, making it harder to monitor them manually. So, automated systems are needed. This change increases the demand for automated systems that detect abnormal events or anomalies, such as road accidents, fighting, snatching, car fires, and explosions in real-time. These systems improve detection accuracy, minimize human error, and make security operations more efficient. In this study, we proposed the Composite Recurrent Bi-Attention (CRBA) model for detecting anomalies in surveillance videos. The CRBA model combines DenseNet201 for robust spatial feature extraction with BiLSTM networks that capture temporal dependencies across video frames. A multi-attention mechanism was also incorporated to direct the model’s focus to critical spatiotemporal regions. This improves the system’s ability to distinguish between normal and abnormal behaviors. By integrating these methodologies, the CRBA model improves the detection and classification of anomalies in surveillance videos, effectively addressing both spatial and temporal challenges. Experimental assessments demonstrate that the CRBA model achieves high accuracy on both the University of Central Florida (UCF) and the newly developed Road Anomaly Dataset (RAD). This model enhances detection accuracy while also improving resource efficiency and minimizing response times in critical situations. These advantages make it an invaluable tool for public safety and security operations, where rapid and accurate responses are needed for maintaining safety.

## 1. Introduction

The rapid growth of surveillance camera deployment in public and private spaces has significantly enhanced public safety and security in urban areas [1]. Closed-Circuit Television (CCTV) systems play a vital role in monitoring public areas and are key components in safety, traffic control, and urban management. Despite advancements in surveillance technologies, the manual observation of video feeds continues to be a primary method for analysis [2]. This process is time-consuming, prone to errors, and costly due to human involvement in monitoring extensive video footage. The constant need for human attention can cause fatigue, lowered concentration, and overlooked important events, making manual monitoring ineffective, especially in large-scale surveillance systems [3]. CCTV cameras capture unexpected events or anomalies, such as accidents, fights, thefts, or fires which are difficult to detect and respond to quickly. Identifying these anomalies is important for improving public safety, speeding up law enforcement responses, and preventing harm to people [4]. However, detecting these abnormal events in surveillance footage demands advanced methods that surpass basic monitoring. This study specifically addresses the challenge of automatically detecting abnormal events in video surveillance systems using cutting-edge technology. The study aims to enhance the effectiveness of CCTV systems [5,6]. Anomalous refers to the identification of unusual or irregular patterns in video data that deviate from typical or expected behavior [7].

An automated Anomaly Detection (AD) system can alert authorities to critical incidents in real-time efficiently, improving public safety and reducing the reliance on human oversight [8]. Anomaly detection (AD) might flag unusual activity in a specific area, while situation recognition helps us understand the context and determine the appropriate response [9]. Despite remarkable advancements in camera movement, occlusions, variable illumination, and the suspicious behavior of humans remain challenging. Anomaly detection (AD) is used in various fields, such as detecting credit card fraud [10], identifying faults [11], diagnosing medical conditions [12], analyzing human behavior [13], and detecting threats [14] in computer networks.

In recent years, Convolutional Neural Networks (CNN) have been widely recognized in deep learning for their ability to extract features from image data by learning multiple levels of feature representation [15]. With their sophisticated categorization and architecture, CNNs may be used for a wide variety of applications because of their ability to capture fine features in high-dimensional datasets [16]. CNNs are primarily applied in computer vision tasks like facial recognition, image classification, object detection, and even in Natural Language Processing (NLP) [17,18].

CNNs are effective for many types of data but not ideal for time series data [19]. To overcome this limitation, Recurrent Neural Networks (RNNs) were introduced to process information in both forward and backward directions, making RNNs well-suited for capturing temporal patterns in time series data [20]. As a result, the model may generate predictions at each time step based on historical data. However, as the data passes through multiple time steps, earlier information can become less impactful. To address this limitation, advanced RNN variants like long short-term memory (LSTM) networks have been introduced, which are more effective at preserving information over longer durations. LSTM was made to identify order dependencies in jobs involving sequence prediction [21]. Long-term dependencies may be learned because of the utilization of memory cells which can hold states across time. Simultaneously, LSTM is adept at modeling temporal dependencies in video sequences. Bidirectional long short-term memory (Bi-LSTM) builds upon the standard LSTM architecture to overcome its limitations by incorporating both the past and future context in sequence modeling tasks. While traditional LSTM models only process input data in a single forward direction, Bi-LSTM enhances this by training the model to analyze data in both forward and backward directions. This dual approach allows for a more comprehensive understanding of the sequence. Recognizing abnormalities in videos is a challenging task because video data encompasses both spatial (where things are) and temporal (how things change over time) dimensions. To address this, it is essential to identify key features in each frame and understand how they relate to adjacent frames over time. Spatiotemporal features are progressively enhanced by combining handcrafted elements with motion-based features to improve accuracy [22]. Training a large capacity of data is a significant problem for researchers. To address this, transfer learning (TL) has been introduced as a highly advantageous and popular model using pre-trained weights on the ImageNet dataset. Most deep learning models based on the transfer learning approach have been used to detect anomalies [23]. We propose an innovative attention-based anomaly detection system designed to analyze spatiotemporal features and focus on long-term patterns to identify anomalies in video frames. This system integrates a CNN equipped with attention modules to improve feature learning, alongside a Bi-LSTM network that leverages attention weights to emphasize crucial details for motion detection. The combination of a CNN for spatial feature extraction with Bi-LSTM processing improves the accuracy of anomaly detection. Our approach depends on frame anomaly detection as a multi-scene problem within supervised learning, focusing on the University of Central Florida (UCF) Crime dataset [24] and the Road Anomaly Dataset (RAD) [25], which include various normal and abnormal activities. The key contributions of this work are as follows:This study emphasizes the critical need for automated road anomaly detection with high accuracy. The proposed model is based on the Composite Recurrent Bi-Attention (CRBA) model for anomaly detection in video streams. This model effectively captures spatial and temporal features, overcoming the limitations of earlier approaches.The state-of-the-art of this model is the incorporation of the CNN pre-trained model DenseNet201 for spatial feature extraction, multiple layers of Bi-LSTM for acquiring temporal patterns, and an attention mechanism to highlight critical spatiotemporal details in surveillance videos. The attention mechanism in the Bi-LSTM layer helps the model focus on important information and ignore irrelevant data, improving its performance.The benchmark of the research was to create a custom real-world dataset for road anomaly detection (RAD). RAD data included recorded videos and images from surveillance systems in different cities of Pakistan. Furthermore, we enhanced the dataset through association with international partners who contributed videos.The proposed framework demonstrates strong effectiveness in accurately detecting road anomalies, achieving accuracy rates of 92.2% on the RAD and 86.2% on the UCF Crime dataset.

The paper is organized into six sections: Section 2 is a related literature review, Section 3 details the proposed methodology, Section 4 describes the experimental results, and Section 5 covers the conclusions, limitations, and future research directions.

## 2. Literature Review

Detecting anomalous events in surveillance videos has been a major research focus for many years. Nonetheless, it remains a difficult task due to the variability within event categories and the influence of geographical factors. The literature on anomaly detection is divided into two different categories including the traditional approach with features-based models and deep spatiotemporal feature learning.

### 2.1. Traditional Approach: Feature-Based Model

Traditional approaches to methods for anomaly detection typically involve a three-step process. The primary phase focuses on feature extraction using manually crafted descriptors, followed by the encoding of these features through a designated algorithm in the second phase. The final step involves categorizing the encoded data using an appropriate machine-learning model [26,27]. In their study, Zhu et al. [28] introduced a temporal augmented network aimed at learning motion-aware features. This feature on its own achieved competitive results when compared to prior state-of-the-art methods, and its combination with these methods led to notable performance enhancements. Additionally, they incorporated a temporal context into the Multiple-Instance Learning (MIL) ranking model by employing an attention block. The attention weights learned through this process improved the differentiation between anomalous and normal video segments. However, the model encountered difficulties in certain challenging situations such as fast motion, groups of people, and low resolution. PWC-Net is regarded as an efficient alternative to FlowNet2, offering comparable performance while operating at a much faster speed for optical flow motion estimation. Working with deep learning models on extensive anomaly datasets in a fully supervised setting presents challenges due to the high computational demands needed to handle both spatiotemporal and motion features. Despite these challenges, a well-trained model in such a setup can enhance the generalization capabilities that enable it to detect various anomalous events and deliver class-specific alerts for real-time autonomous surveillance systems. In certain instances, classifiers’ performance can significantly improve in supervised environments when background elements in video scenes like roads and vehicles in traffic accident detection are clearly defined [29]. Anomalous patterns are identified using training videos with frame-level temporal annotations. A widely adopted approach combines prior geometric knowledge through object detection or semantic segmentation, often further refined using additional supervision from publicly available datasets. In computer vision, two key feature extraction strategies dominate local feature-based and global feature-based methods. Local feature-based approaches capture features as distinct patches, key points, and gesture-based information aligned with specific task-relevant aspects. On the other hand, global features encompass broader regions of interest, typically represented through background subtraction and object-tracking techniques [30]. In the machine learning approach, researchers have developed systems that rely on handcrafted features. These feature extractors are often specifically designed for certain datasets, making them highly specialized and not suitable for general-purpose feature learning [31]. This specialization limits their use beyond the datasets for which they were primarily developed. Some researchers have introduced keyframe-based methods to reduce the processing time [32] to improve efficiency. For instance, Yasin et al. [33] exhibited a technique for selecting keyframes in video sequences, which was applied to human activity recognition. Similarly, Zhao et al. [34] created a human activity recognition algorithm based on multi-feature fusion utilizing keyframes and conventional machine learning approaches. Traditional machine learning algorithms have come a long way but have been limited by human intuition and encounter challenges including long processing times, labor-intensive jobs, and feature engineering complexity.

### 2.2. Deep Spatial–Temporal Features-Based Model

Deep learning offers a new approach compared to the traditional approach. It uses an end-to-end model that learns and represents complex visual features all at once during classification [35]. CNNs adjust their parameters dynamically based on the input data, using convolutional operations to identify the most relevant features [36] from end to end. Some researchers have developed techniques using 3D filters for analyzing video frames [37]. These methods have proven more effective for tasks such as action recognition, object tracking, and video retrieval. CNNs perform better than traditional handcrafted feature approaches. The introduction of 2D and 3D deep learning methods has led to more efficient recognition [38]. The shift toward 3D feature extraction demonstrates how deep learning models can adapt to diverse visual data and discern intricate patterns in dynamic sequences. However, these methods often face challenges in capturing comprehensive patterns between frames due to shared parameter limitations over time and their emphasis on localized input regions [39]. To address this, Recurrent Neural Networks (RNNs) present different hidden units for handling sequential and temporal data. Spatiotemporal features have proven to be highly effective for understanding video events and play a crucial role in video classification to capture both foreground and background dynamics [40]. In this study, Peng et al. [41] proposed a model to estimate the spatiotemporal and motion relationships between frames, rather than treating them merely as a sequence of images like videos. Additionally, a new approach involves a clustering-based convolutional autoencoder that uses spatial and motion autoencoders to generate optical flow [42]. Another advancement introduced by Chang et al. [43] is a variance attention module which improves the learning of spatial and motion features within the convolutional autoencoder framework. In this study [44], a recurrent encoder-decoder network was suggested for anomaly detection by learning both global body movements and local postures from dynamic skeletal data. Additionally, an evolving graph network was introduced. In their study, Yang et al. [45] represented individuals as nodes and their movements as node–edge relationships, utilizing optical flow to detect anomalies in dense crowds. Furthermore, the Spatial-Temporal Action Translation (STAT) network has been developed to reconstruct frames using optical flow features, which assists in identifying violent actions [46]. In another study, Jiang et al. [47] developed the LSTMDT model based on LSTM. This model tracks changes in traffic conditions before collisions and captures trends across various periods. The LSTMDT model has shown better performance compared to traditional machine learning classifiers for traffic accidents. In their study, Kang et al. [48] introduced the Vision Transformer Traffic Accident (ViT-TA) classifier, which analyzes traffic accidents using first-person video footage to enhance the safety of autonomous vehicles. However, several issues have been found. Simply sharing parameters over time is not enough to fully understand the relationships between input samples. In their study, Zaheer et al. [49] proposed a model based on a convolutional 3D (C3D) network to improve feature extraction from videos containing anomalies. They utilized the binary clustering of spatiotemporal features to create simulated labels, which facilitated self-supervised training and reduced noise in the labels of anomalous videos. The C3D model was evaluated on well-known real-world anomaly detection datasets such as UCF Crime and UCSD Ped2, showing better performance compared to existing top methods. However, the study observed that the processing time increased for larger video segments. Similarly, Singh et al. [50] proposed a framework for automatic traffic accident classification using a denoising autoencoder, which avoided traditional deep feature extraction from raw pixels and instead calculated the accident probability from depth representations. Following this, Vishnu et al. [51] employed hybrid stop filtering to reduce noise in traffic accident videos and used SVM for vehicle tracking to detect accidents based on the traffic density and vehicle statistics. However, this approach requires significant computational resources. These models primarily focus on deep features for classification but often overlook the connections between accident-related objects and other essential features. Previous research usually treated spatial and temporal features separately when detecting anomalies, often employing CNNs for spatial analysis and RNNs for temporal analysis independently. Several methods for anomaly detection using deep learning have been introduced, but challenges remain in effectively capturing both spatial and temporal features. Moreover, the lack of road-specific anomaly datasets hampers progress in this field. Future research could benefit from tackling these challenges, particularly by enhancing the integration of spatial and temporal feature learning and developing more detailed road anomaly datasets.

## 3. Model Architecture

The proposed model’s efficiency is significantly enhanced through the integration of various advanced techniques, as depicted in Figure 1. First, it captures a set of frames from the input video and processes these frames to extract significant features and then optimizes feature extraction by employing convolutional layers, MaxPool filters, batch normalization, and dropout layers. The output from these layers, structured as multidimensional matrices, is flattened into a one-dimensional form and then passed to the dense layer. The use of ReLU was deliberately chosen to improve computation and softmax activation functions improved the prediction accuracy. We utilized a pre-trained CNN model called DenseNet201, which played a crucial role in our approach. This selection not only improved the computational efficiency but also leveraged the model’s strengths. Moreover, we employed a multilayer BiLSTM classification model with an attention layer to further enhance the efficiency.

We refined the hierarchical features extracted from the DenseNet201 layer. This hierarchical modification improved the model’s ability to distinguish features by capturing sequential dependencies. The BiLSTM paired with the attention layer block is highly effective in identifying long-term temporal patterns, offering a detailed understanding of dynamic actions. The model adjusts dynamically to various temporal contexts to ensure a reliable performance across diverse scenarios. Algorithm 1 presents each step performed throughout the study. In the first step, the video was first processed to extract frames, creating a dataset. In the second step, the DenseNet201 pre-trained model was used to automatically extract features from each frame. A pooling layer was added to support the transfer learning process and extract the key features. In the third step, the output was sent to the multilayer BiLSTM associated with the attention layers, which received input from the spatial and temporal feature extraction after extracting the key features. There were multiple residual attention BiLSTM blocks assembled into a stack. Each BiLSTM layer integrated with an attention layer moved the output of the upcoming preceding block to the next BILSTM block, and the softmax layer was used to categorize the road anomalies. In the fourth step, we trained the model and evaluated the validation results, and step five involved presenting the results in terms of accuracy and loss graphs. In step six, real-time testing was performed, enabling users to upload any road anomaly video for the model to classify.
**Algorithm 1.** Proposed model for classification.**Input:** Number of videos. **Output:** Classification of road anomaly. **Step 1: Preprocessing**
  while input_videos ≠ last_input_video     sequence of frames ← video to image conversion     preprocess_video.append(sequence of frames) **Step 2: Feature extraction using pre-trained models**
   video features ← load models **Step 3: Classification**
   class models ← deep_bi_lstm(video features) **Step 4: Make predictions**
   while i ≠ num_epochs do      //for training      while Batch_video_features not equal number-batch-training do       predicted_class ← class models(Batch_video_features)       loss ← Criterion(predicted class)       update loss ← loss.background(), optimizer.step()    while Batch-video-features ≠ number-batch-testing do      predicted_class ← class-models(Batch_video_features)      output ← confusion-matrix(predicted class, ground truth table) **Step 5: Show results**
   print (“Performance”, accuracy) **Step 6: Recorded video testing**
   anomaly-test-video ← Recorded-video    predicted-class-name ← Class models(anomaly)    print(“Predicted”, Anomaly name)

### 3.1. DenseNet201

DenseNet201 is a CNN-based deep learning model. Using DenseNet201 for feature extraction leverages the model’s training on the ImageNet dataset. In this framework, DenseNet201 is a pre-trained CNN model. It is used for spatial feature extraction. It connects each layer to every preceding layer within dense blocks, enabling extensive feature reuse and making the model more compact and resistant to overfitting. This design also allows for the direct supervision of each layer through shortcut paths, providing implicit deep supervision, which is ideal for detailed tasks like per-pixel prediction.

The essential structure of DenseNet201 in our test was made up of four dense blocks, each with four layers. The move-down layers, which consisted of a 1 × 1 convolution, and 2 × 2 max-pooling layers were laid in between these dense blocks as shown in Figure 2.
X_l_ = H_l_([X_0_, X_1_, …, X_l−1_])(1)

In Equation (1), [X_0_, X_1_, …, X_l−1_] is a single tensor structured by the sequence of the previous layers of output the feature maps. This architecture uses a loss function to directly supervise each layer by using the sequence. In this design, the function is a combination of four processes that are regarded as one layer: dropout, 3 × 3 convolution, batch normalization, and leaky rectified linear unit (LReLU).

### 3.2. Long Short-Term Memory (LSTM)

The LSTM network improves traditional Recurrent Neural Networks (RNNs) by adding memory capabilities. RNNs often have trouble handling long-term dependencies and face problems like vanishing or exploding gradients [53]. LSTM overcomes these by incorporating three gating units and a state update mechanism in each cell, allowing for dynamic internal circulation weights and variable integral scaling over time while keeping network parameters constant. This design significantly improves the model’s generalization capabilities.

The basic structure of the LSTM model is shown in Figure 3. The LSTM model performs better in processing sequential data because of its improved long-term and short-term memory capabilities. The input gate can accept or reject input features, generating memory cells. In the next layer, the input gate can either pass the memory cell state to the neuron or discard it. The forget gate regulates cell state changes, determining whether to retain current characteristics or discard past ones.

### 3.3. Bidirectional Long Short-Term Memory (Bi-LSTM)

The Bi-LSTM model effectively captures temporal dependencies in sequential data enhancement and uses input information by accurately tracking its flow. It comprises forward and backward modules. Each is constructed with bidirectional long short-term memory networks featuring distinct architectures [54]. These modules share a common output layer, allowing each unit to access comprehensive information from both past and future inputs simultaneously. Each operation in the model is linked to a weight, w. During Bi-LSTM operation, data flows through both forward and backward hidden states, resulting in a hidden layer output that incorporates bidirectional temporal processing.

Figure 4 shows *x*_(*t*−1)_, *x_t_*, and *x*_(*t*+1)_ as the input data at times (*t* − 1), (*t*), and (*t* + 1), respectively, while *h*_(*t*−1)_, *h_t_*, and *h*_(*t*+1)_ represent the hidden states at these respective times in the LSTM. The *o*_(*t*−1)_, *o_t_*, and *o*_(*t*+1)_ represent the corresponding output data. *w*_1_, *w*_2_, …, and *w*_6_ indicate the weight of each layer. The status update of the hidden layer of forward and backward LSTM and the final output process of BiLSTM are shown in Equations (2)–(4).
(2)$$h_{t}=f_{1}\left({w_{1}x}_{t}+w_{2}h_{t-1}\right)$$
(3)$${h^{\prime }}_{t}=f_{2}\left({w_{3}x}_{t}+w_{4}{h^{\prime }}_{t+1}\right)$$
(4)$$o_{t}=f_{3}\left({w_{5}h}_{t}+w_{6}{h^{\prime }}_{t}\right)$$
where $f_{1}$, $f_{2}$, and $f_{3}$ present activation functions between different layers, respectively.
(5)$$f_{t}=\sigma (W_{\dot{f}}\left[h_{t-1}x_{t}\right]+b_{f})$$


(6)
$$v_{t}=tanh(W_{\dot{c}\;}\left[h_{t-1}x_{t}\right]+b_{v})$$


Determine the input state $i_{t}$:(7)$$i_{t}=\sigma (W_{\dot{c}}\;\cdot \left[h_{t-1}x_{t}\right]+b_{v})$$

Calculate the cell state $C_{t}$:(8)$$C_{t}=f_{t\;}\cdot C_{t-1}+i_{t\;}\cdot v_{t\;}$$

The output gate $o_{t}$ and the hidden state $h_{t}$ in the current layer:(9)$$o_{t}=\sigma (W_{o\;}\left[h_{t-1}x_{t}\right]+b_{o})$$
(10)$$h_{t}=o_{t\;}tanh(C_{t})$$

In this context, *W* and *b* represent the weights and biases of the training matrix, respectively. The symbol *σ* denotes a nonlinear activation function that outputs values.

Within the range of [0, 1], the hidden layer unit is denoted by *h*, while $f_{t}$ represents the forgotten gate unit. The cell state unit is indicated by $v_{t}$. The update gate unit, also known as the input unit, is denoted by $i_{t}$. The cell state is represented by $C_{t},$ and $o_{t}$ is the output gate, which synchronizes and outputs information from the previous unit. Bi-LSTM procedures sequence information in both forward and backward directions and integrate this material into the current output layer. Bi-LSTM enhances prediction accuracy by utilizing information from both previous and current data. The hidden layer combines results from forward and backward processing using cascading vectors. The spatiotemporal feature extraction module serves as input for the attention module. This is achieved by stacking multiple residual attention Bi-LSTM blocks connected in sequence, with the output from each block being passed to the initial Bi-LSTM of the following block. Let $c_{t}^{\left(i\right)}$ be the temporal features vector made by the Bi-LSTM concerning *t*, time. The attention layer produces a background vector, *v^t^*, for $c_{t}^{\left(i\right)}$ at time *t*. This vector is produced by applying attention weights, $w_{t}^{\left(i\right)}$, using Equation (11) to calculate the situational vector.
(11)$$v_{t}=\sum_{i=1}^{N}w_{t}^{\left(i\right)}c_{t}^{\left(i\right)}$$

An activation function is applied to the hidden state $H_{t}$ of the first Bi-LSTM layer to compute the relevance score *r*(*i*) at time *t*, as described in Equation (12).
(12)$$r_{t}^{\left(i\right)}=tanh(WH_{t}+b)$$

The relevance of feature *i* at time t is represented by $r_{t}^{\left(i\right)}$. *W* and *b* denote the model weight and bias, respectively. These parameters are adjustable, which impacts the model’s performance. The activation function utilized is the hyperbolic tangent, tanh().

The Bi-LSTM network is well suited for capturing long-term dependencies, but it can struggle to pinpoint the exact input elements crucial for precise classification. This challenge can be mitigated by integrating an attention mechanism. The output vectors *f*_1_, *f*_2_, *f*_3_, …, *f*_*n*_ from the Bi-LSTM network are fed into the attention layer. Here, data information is translated into the vectors *v*_1_, *v*_2_, *v*_3_, …, *v*_*n*_ by the attention encoder block, and then the context vectors *C*_1_, *C*_2_, *C*_3_, …, *C*_*n*_ is computed using the weighted sum of the encoder RNN output, as shown in Equation (13).
(13)$$C_{t}=\sum_{t=0}^{n}f_{t}v_{t}$$

The attention module computes the attention weight $A_{t}^{\left(i\right)}$ at time *t*, as detailed in Equation (14).
(14)$$A_{t}^{\left(i\right)}=\frac{\exp\left(w_{t}^{\left(i\right)}r_{t}^{\left(i\right)}\right)}{\sum_{j=1}^{N}\exp\left(w_{t}^{\left(i\right)}r_{t}^{\left(i\right)}\right)}$$

Here, $w_{t}^{\left(i\right)}$ represents the model weight learned for extracted feature i at time t and collects high-level sequence information in our method by using a stacked RNN with many LSTM layers. Before being sent to the output, data in LSTM are often processed through one layer. To tackle temporal sequence problems, data are analyzed across multiple layers. In this architecture, the hidden state from one layer is passed to the next, forming a hierarchical arrangement where all layer’s work together to process the input effectively. Both communication vector $C_{t-1}$ and the attention score $a_{t}$ are utilized in the encoding. The attention scores are computed using Equation (15), which shows the softmax of the attention score at time t. Equation (16) represents the prior cell state vector as $D_{t-1}$.
(15)$$o_{t}=f_{att}\left(C_{t-1}D_{t-1}\right)$$


(16)
$$softmax\left(a_{t}\right)=\frac{\exp\left(o_{t}\right)}{\Sigma_{t=0}^{n}\exp\left(o_{t}\right)}$$


At each time *t*, the attention layer’s output is determined using the perspective vectors *C_t_*, the output $o_{t}$ from the previous time step *C_t_*_−1_, and the previous cell state vector *D_t_*_−1_.

### 3.4. Attention Mechanism

Attention-based models are a type of deep learning architecture that allows the model to focus on the key features of the input data while disregarding irrelevant information. Unlike traditional neural networks where all input features are treated equally, attention mechanisms enable the model to assign different levels of importance to each feature. This selective attention enables the model to focus on the most informative frames or regions within a frame, which is especially helpful in video-based anomaly identification. As a result, attention-based models can improve accuracy and reduce computational complexity. The basic arrangement of the attention mechanism is shown in Figure 5.

Detecting anomalies in video surveillance is difficult because of the changing environment, different visual features, and the challenge of recognizing patterns over time. Road anomalies like accidents, fights, car fires, and snatching often go unnoticed because there aren’t enough advanced models or specialized datasets. This challenge is made worse by the imbalance and lack of road-specific annotated datasets, which are essential to train and test successful detection algorithms. The CRBA model solves these problems by using CNNs, Bi-LSTM, and attention mechanisms. It applies attention to both spatial features (using CNNs) and temporal patterns (using Bi-LSTM), helping to prioritize important information at multiple levels of the attention model. This comprehensive approach improves the overall performance in detecting road anomalies in surveillance videos.

## 4. Experimental Evaluation and Performance

In this section, we discuss the datasets and experiment design that comprise the base of our study. Following this, a comparative analysis is conducted between the proposed model and several established state-of-the-art approaches in anomaly detection, highlighting the strengths and improvements achieved in feature learning and detection accuracy. Furthermore, we explain the outcomes of the systematic evaluation of various network components via ablation experiments.

### 4.1. RAD

The RAD was compiled from multiple sources to capture a broad spectrum of road anomalies. Videos and images were recorded using mobile cameras and surveillance systems in different cities of Pakistan. To the best of our knowledge, this is the first publicly available road anomaly dataset sourced from a South Asian country. The dataset covers a wide range of instances, showcasing diversity in size, shape, and environmental contexts. This dataset is highly valuable for advancing intelligent transportation and surveillance systems. The videos captured the trajectory angles of vehicles, showing their paths and directions. The dataset consists of videos and images of five distinct types of road abnormalities. The videos are available in the resolutions of 1920 × 1080 @25 frames per second, 848 × 480 @30 frames per second, and 640 × 480 @24 frames per second, which were used to extract frames [25]. The RAD consists of four real-world road anomalies: road accidents, car fires, fighting, and snatching (at gunpoint). Figure 6 illustrates the sample of the RAD. Table 1 offers a summary of the RAD and Figure 7, the distribution of the dataset.

### 4.2. UCF Crime Dataset

The UCF dataset was developed by the University of Central Florida and was utilized to assess the proposed approach. It is a large and realistic dataset with 13 types of real-world abnormal events like abuse, arrests, arson, assaults, accidents, burglary, explosions, fighting, robbery, shooting, stealing, shoplifting, and vandalism, as shown in Figure 8 [55]. Many videos in this dataset are long and contain different scenes, making it difficult to detect and recognize abnormal events. Figure 9 shows its distribution. UCF Crime has been divided into four major categories (4MajCat): road accidents, fighting, explosions, and stealing. Table 2 summarizes the number of videos for training from the 4 major categories of the UCF Crime dataset. Road accidents refer to an incident involving a dangerous interaction between humans and vehicles, often leading to serious injury or death. Such incidents are regarded as suspicious. Fighting involves physical violence between two or more individuals where the intent is to cause harm, and any explosion occurring in a public space fall under this category because of its potential for harm. Theft is considered suspicious, especially when it presents a significant threat to individuals or public safety. 

### 4.3. Performance Measure

The model was evaluated by using several performance measures, including the Receiver Operating Characteristic (ROC) curve, confusion matrix, accuracy, recall, precision, and F1-score. In anomaly detection (AD) systems, it is crucial to minimize false positives (FPs) and false negatives (FNs) while maximizing true positives (TPs) and true negatives (TNs). True negatives refer to normal instances that are accurately identified, and true positives are anomalies that are correctly detected. On the other hand, false positives are normal instances mistakenly classified as anomalies, while false negatives are anomalies incorrectly classified as normal. Each assessment metric is computed as follows.
(17)$$Accuracy=\frac{TP+TN}{TP+FP+TN+FN}$$


(18)
$$Precision=\frac{TP}{TP+FP}$$



(19)
$$Recall=\frac{TP}{TP+FN}$$



(20)
$$F1\text{-}Score=2\;\times \;\frac{Precision\;\times \;Recall}{Precision+Recall}$$


### 4.4. Experimental Setup

This study was carried out on the Intel Core i7 CPU, 10th generation, with 16 GB RAM and an 8 GB NVIDIA RTX 3090Ti GPU (Graphics Processing Unit) in the presence of a Windows operating system-based machine. CUDA nvJPEG 12.3.3.54 was used to speed up the processing of the GPU. In the context of the programming environment, Python 3.9.1, we used the Tensorflow version 2.2 and Keras version 2.3.1 libraries to perform experiments. Additionally, the proposed approach was evaluated on two benchmark datasets, the UCF dataset and the Road Anomaly Dataset (RAD), both publicly available for researchers.

Each video was represented by 25 evenly spaced frames selected across its entire duration, ensuring efficient memory usage by the classification models without compromising their performance. Data augmentation approaches such as flipping, zooming, and rotating were employed to expand the dataset. The pixel values of each frame were normalized by dividing them by 255 and scaling them to a range between 0 and 1. The dataset was then shuffled and divided into training (70%), validation (20%), and testing (10%) sets for the modeling process. We configured the model with a batch size of 32, a learning rate of 1 × 10⁻⁴, and 50 training epochs, using Glorot Uniform initialization for the weights and an Adam optimizer. Each training batch consisted of 10 videos to optimize the model’s performance. Our proposed model adopted the Adam optimizer, which stands for Adaptive Moment Estimation, which is a well-established optimization technique in both machine learning and deep learning. The Adam optimizer with default settings was used to oversee the optimization procedure. To address a class imbalance, categorical focal loss was utilized, allowing the model to focus on the key features of violent videos and improve its detection performance. Early stopping was used to mitigate the risk of overfitting.

### 4.5. UCF Dataset Results

This section presents the results of the proposed model on four major threats to public safety: accidents, explosions, fighting, and theft, using the UCF dataset. The dataset was characterized by imbalanced instances, with each activity occurring in various scenarios and locations [56]. To estimate the performance of the proposed DenseNet201 + multilayer (BILSTM + attention) model, we compared it against several other models including DenseNet201, DenseNet169, DenseNet169–BiLSTM, DenseNet201–BiLSTM, and the proposed CRBA model, presented in Table 3.

Figure 10a delivers insight into the training and validation accuracies on the UCF dataset. Figure 10b illustrates the clear changes in the training and validation loss on the UCF dataset. Figure 10c exhibits the confusion matrix to highlight the class-specific accuracy. Figure 10d presents the ROC curve, demonstrating how the model balances true positive and false positive rates for the UCF dataset.

The proposed model was compared with five prior studies presented in Table 4, including 3D CNNs [57], InceptionV3 + LSTM [58], residual attention LSTM [59], InceptionV3 + BiLSTM [60], and InceptionV4 + BiLSTM + attention [61]. Our proposed model outperforms these with accuracy improvements of 10.5%, 11.63%, 7.77%, 5.19%, and 24.16 percent, respectively.

### 4.6. RAD Results

The Road Anomaly Dataset (RAD) includes videos featuring four distinct types of real-world anomalies. This dataset presents a realistic and complex challenge for anomaly detection, as it encompasses a broad range of conditions, such as diverse camera distances, object sizes, and lighting environments. To evaluate the performance of the proposed CRBA model, we compared it against several other models, including DenseNet201, DenseNet169, DenseNet169–BiLSTM, and DenseNet201–BiLSTM, as shown in Table 5. Figure 11a illustrates the increasing curve representing the training and validation accuracies on the RAD dataset. Figure 11b demonstrates the smooth decline in both testing and training loss on the RAD dataset. Figure 11c exhibits the confusion matrix to highlight the class-specific accuracy. Figure 11d presents the ROC curve, demonstrating how the model balances true positive and false positive rates in the RAD. To assess the performance of the proposed model, we carried out an extensive series of simulations on the two datasets. These experiments involved testing several model variations, such as DenseNet169, DenseNet201, DenseNet169–BiLSTM, and DenseNet201–BiLSTM. A summary of the model’s accuracy across the UCF dataset and RAD is provided in Table 6. In comparing the DenseNet169 and DenseNet201 models, DenseNet201 is proficient at capturing key features, while DenseNet169 is less effective at identifying key features. However, since the data include time series elements, incorporating BiLSTM offers only a slight enhancement in performance, leading to a minimal overall improvement. When evaluating DenseNet201 against the DenseNet201–BiLSTM–attention model, the combination of DenseNet201, BiLSTM, and attention mechanisms shows more efficiency. DenseNet201 extracts local features, BiLSTM identifies temporal patterns, and the attention mechanism focuses on crucial knowledge. This integration allows for more thorough feature extraction and processing, optimizing the use of data features and significantly boosting the model’s performance. An independent test is crucial for assessing a model’s generalization ability by evaluating its performance on previously unseen data. This study conducted an independent evaluation of the proposed multi-task classification model to gauge its effectiveness with novel scenarios and new information. The proposed model, trained on the UCF dataset and the RAD, both containing anomaly instances, was subsequently tested on a video dataset featuring both normal and anomalous scenes, as depicted in Figure 12. This evaluation measured the model’s capacity to generalize learned patterns to unfamiliar data. 

UCF Crime dataset: The proposed DenseNet201 + multilayer (BiLSTM + attention) model showed a significant improvement in detecting the four critical anomalies, accidents, explosions, fighting, and stealing. The model achieved an accuracy of 86.2%. This performance boost highlights the advantage of the multi-attention layer in handling complex spatiotemporal features and improving anomaly detection in imbalanced datasets.

Table 7 shows the precision, recall, F1-score, and accuracy for each of the four classes in both the UCF dataset and RAD. This per-class breakdown provides a more detailed evaluation, allowing us to see how well the model performs on each specific type of anomaly. Additionally, we include a brief analysis to highlight differences in the performance across classes. Detection rates are higher for “car fires” and “explosions” because these events have distinct visual signs, such as flames or smoke, which make them easier to identify. In contrast, “snatching at gunpoint” and “stealing” are more challenging to detect accurately, as they lack obvious visual indicators, resulting in a slightly lower performance for these types.

RAD: The RAD, the first publicly available road anomaly dataset from South Asia, provides a diverse range of scenarios. The CRBA model outperformed other models, with an F1-score of 92.2%. The improvement is due to the model’s multi-attention mechanism, which enhances its ability to manage various environmental conditions and detect road anomalies with greater accuracy. We performed cross-dataset experiments to test the model’s ability to generalize by training it on one dataset and testing it on another. The results show that the model adapts well to different conditions. The model obtained an accuracy of 85.3%, precision of 83.5%, recall of 84.1%, and F1-score of 83.8% when trained on the UCF dataset and evaluated on the RAD. Similarly, 82.7% accuracy, 81.9% precision, 82.3% recall, and an 82.1% F1-score were obtained with training on the RAD and testing on the UCF dataset. These findings demonstrate the model’s dependability for detecting road anomalies by proving that it consistently performs well across datasets. The slight drop in performance when training on the RAD and testing on the UCF dataset may be due to differences in the scene complexity and types of anomalies between the datasets.

## 5. Conclusions, Limitations, and Future Work

In summary, this paper introduces a deep learning framework for anomaly detection in video surveillance, particularly focused on surveillance videos. The model extracts feature from video frames using pre-trained networks, including DenseNet169, DenseNet201, and BiLSTM with an attention mechanism. This setup enables a thorough and efficient analysis of video data by leveraging the strengths of these networks. BiLSTM, enhanced by multilayer attention, improves the model’s capability to comprehend temporal sequences by processing data bidirectionally. The attention mechanism simultaneously integrates multiple aspects of the input sequence, making the model well suited for handling complex video data. To manage long sequences, the architecture incorporates a global max pooling layer to capture essential features, a dropout layer to prevent overfitting, and a sigmoid function for anomaly detection. A fully connected layer is also used to ensure the accurate classification of anomalies. The proposed framework was tested on multiple publicly available datasets containing various anomalous events. The results indicate that it surpasses current video anomaly detection models, achieving high accuracy and showing promise for real-time use in monitoring systems. The model demonstrated high accuracy and robustness across different anomaly types, achieving significant performance improvements over several baseline models. The evaluation was based on various performance metrics, including precision, recall, F1-score, accuracy, confusion matrix, and ROC (AUC), which provided a holistic view of the model’s capabilities. However, we recognize that other metrics, such as the mean Average Precision (mAP) and Area Under the Precision-Recall Curve (AUC-PR), could provide additional insights into the model performance, especially in scenarios with imbalanced class distributions. We plan to explore these additional metrics in future work to provide a more comprehensive assessment. Additionally, the model required significant computational resources for detection which could be a limitation in an environment with fewer resources. Future research will aim to enhance the model’s ability to perform well in challenging weather conditions and make it more versatile in detecting various types of anomalies.

## Figures and Tables

**Figure 1 sensors-25-00251-f001:**
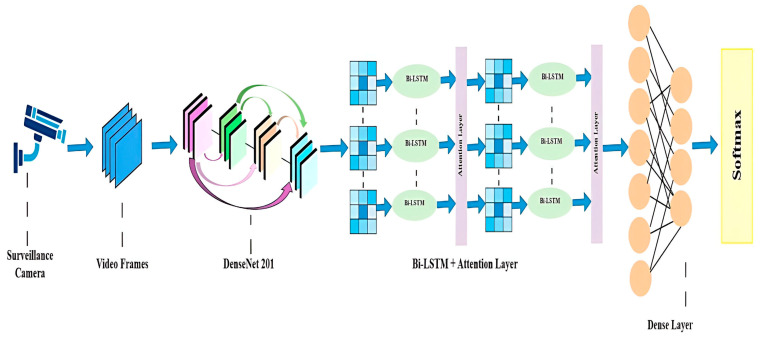
Proposed model architecture.

**Figure 2 sensors-25-00251-f002:**
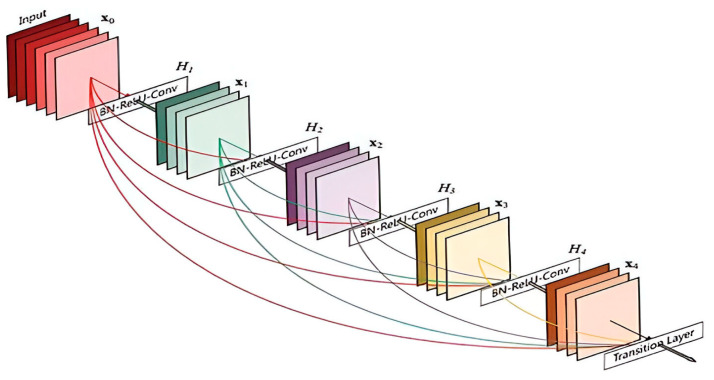
The DenseNet201 model architecture [52].

**Figure 3 sensors-25-00251-f003:**
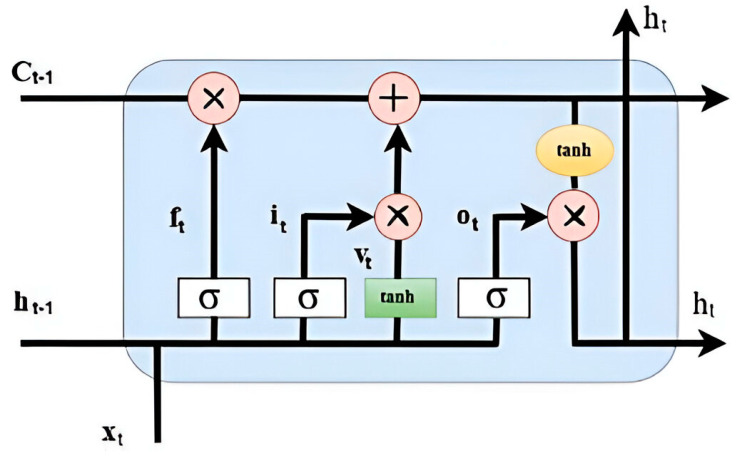
The basic architecture of LSTM.

**Figure 4 sensors-25-00251-f004:**
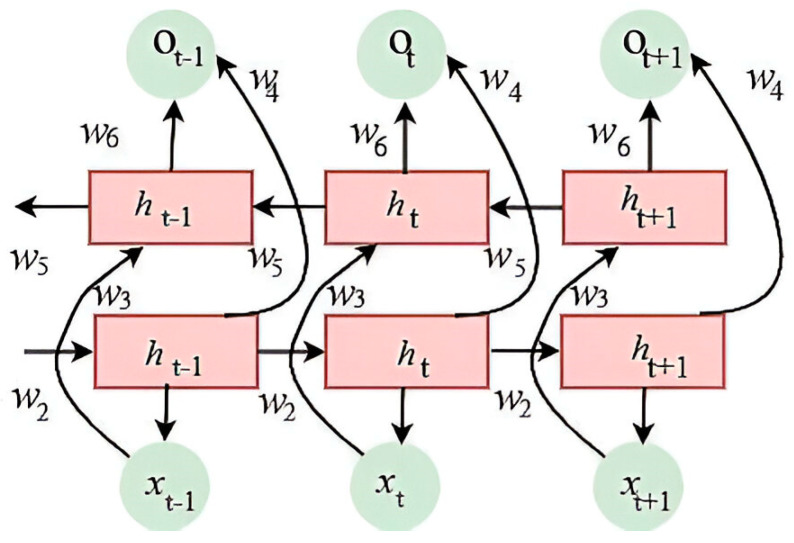
The basic architecture of Bi-LSTM.

**Figure 5 sensors-25-00251-f005:**
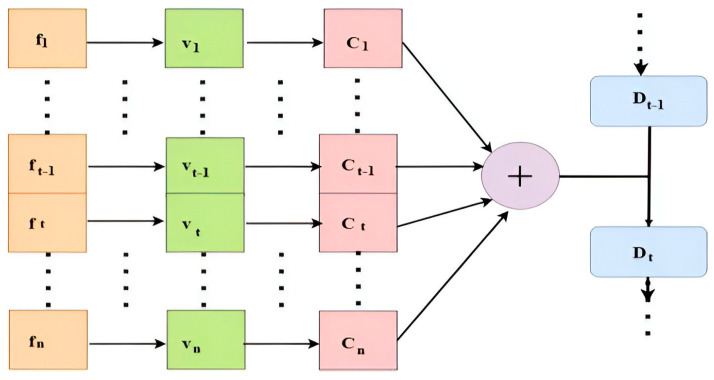
The basic architecture of the attention mechanism.

**Figure 6 sensors-25-00251-f006:**
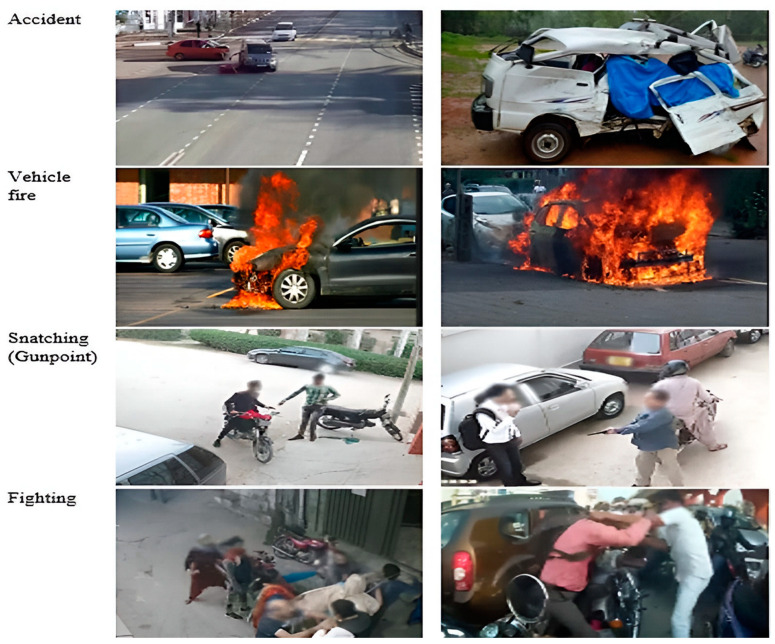
Sample images of the RAD.

**Figure 7 sensors-25-00251-f007:**
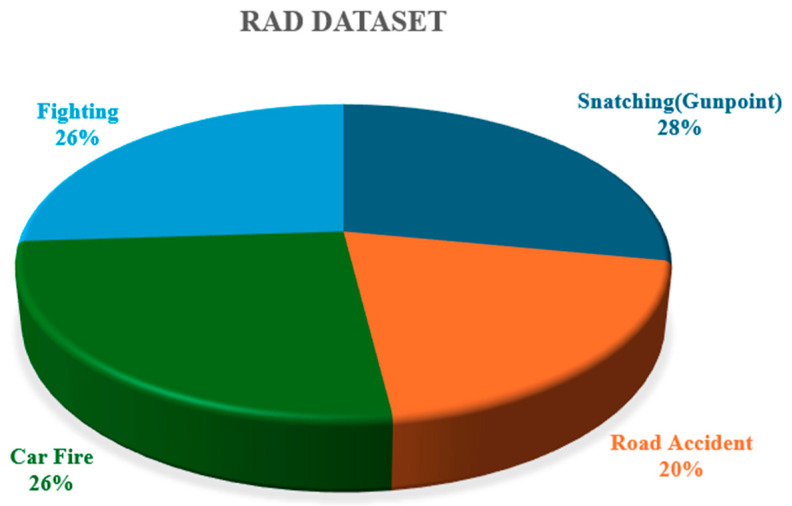
The distribution of the RAD.

**Figure 8 sensors-25-00251-f008:**
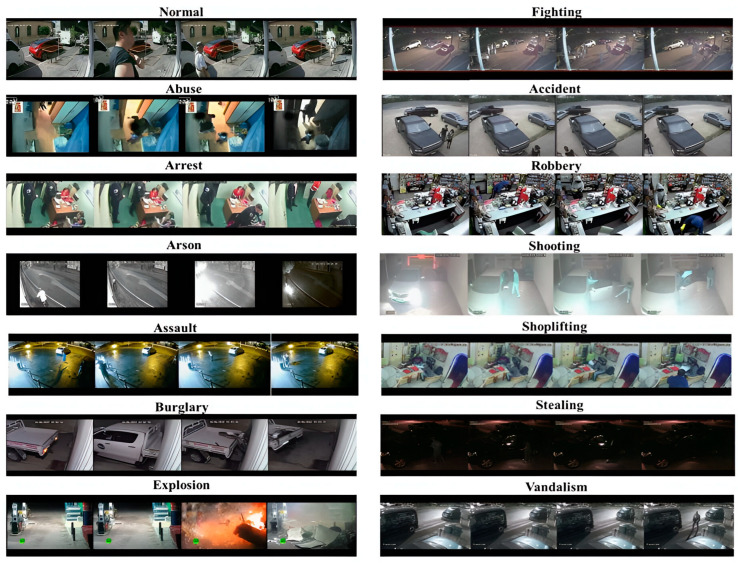
Sample images of the UCF dataset.

**Figure 9 sensors-25-00251-f009:**
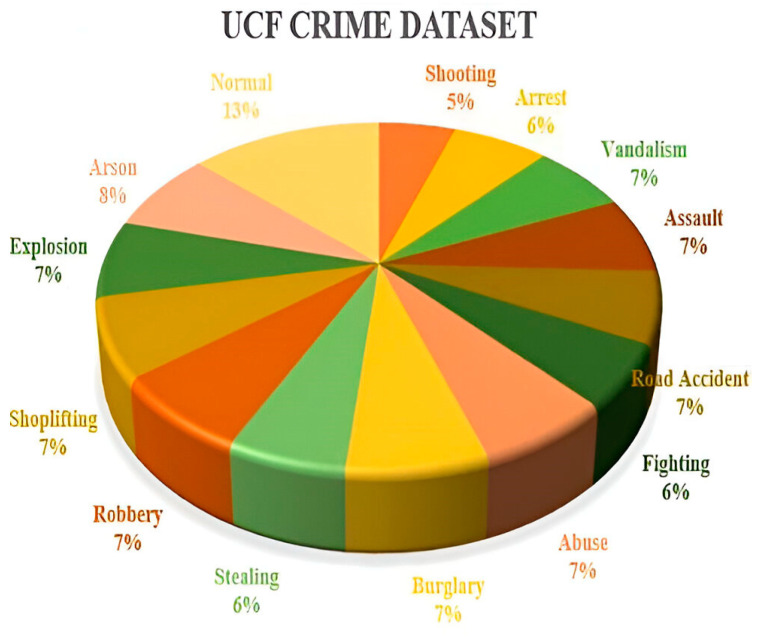
The distribution of the UCF Crime dataset.

**Figure 10 sensors-25-00251-f010:**
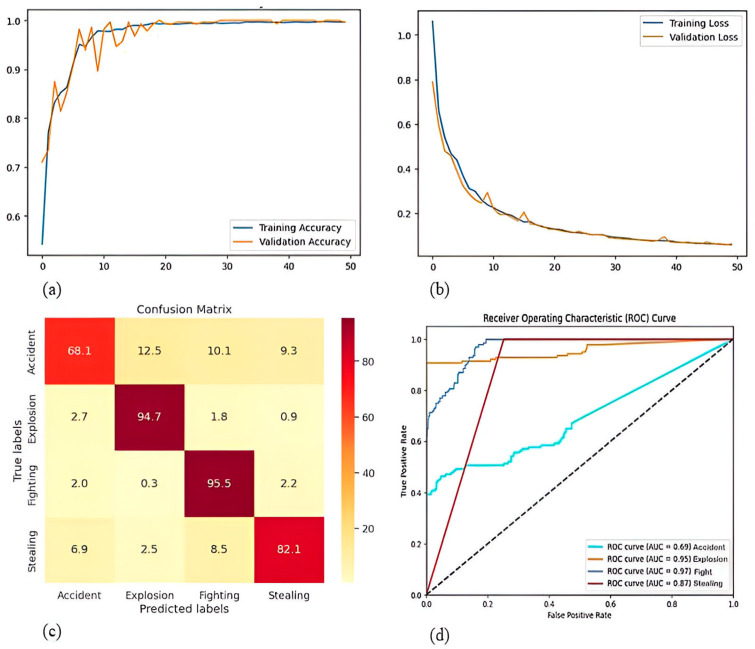
(**a**) Training and validation accuracy curve (**b**) Training and validation loss curve (**c**) Confusion matrix. (**d**) ROC curve of UCF Crime dataset.

**Figure 11 sensors-25-00251-f011:**
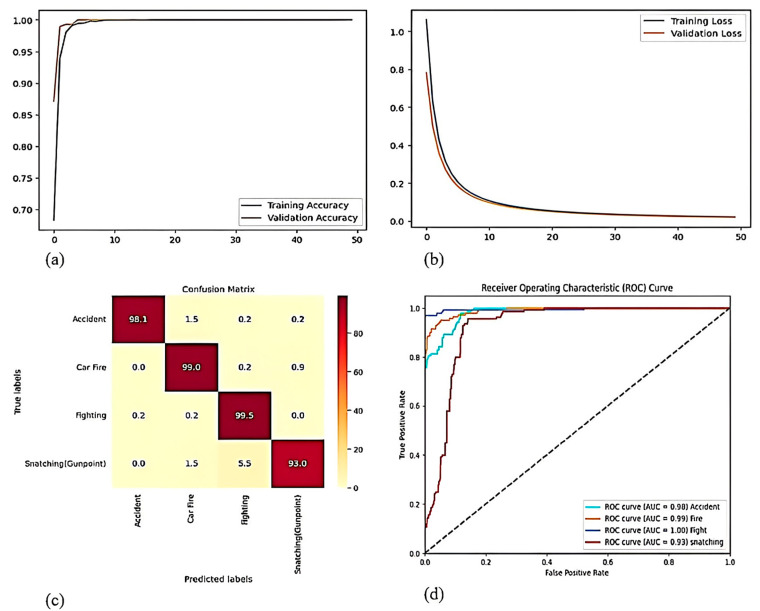
(**a**) Training and validation accuracy curve (**b**) Training and validation loss curve (**c**) Confusion matrix (**d**) ROC curve of Road anomaly dataset.

**Figure 12 sensors-25-00251-f012:**
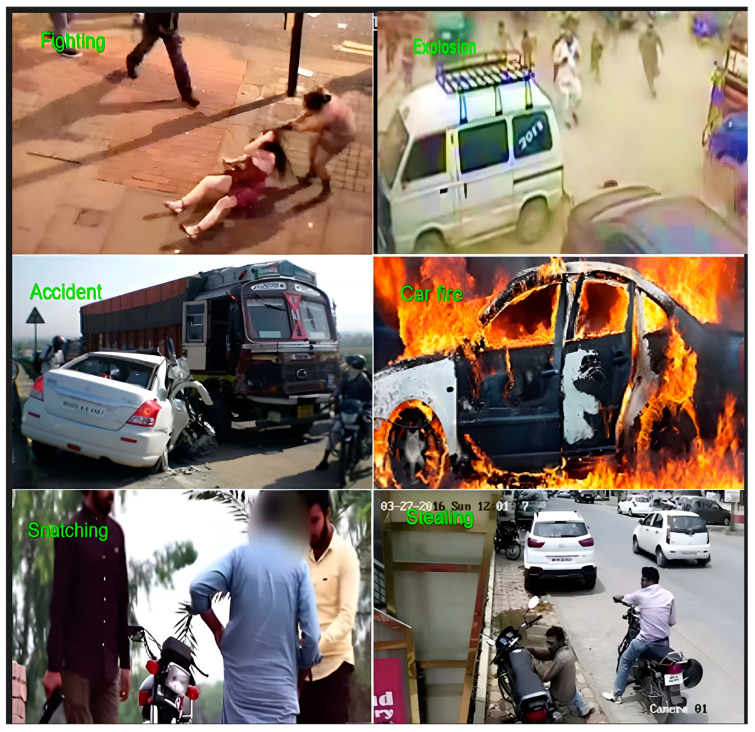
Classification of test videos.

**Table 1 sensors-25-00251-t001:** The main characteristics of the videos included in the RAD.

File Name	Video Length	Resolution (Width × Height) Pixels @ Frames	Lighting Condition	Type of Anomaly
RA03	8 s	640 × 480 @ 25	low	Road accident
RA04	30 s	1280 × 720 @30	high	Road accident
RA05	40 s	640 × 480 @30	high	Road accident
VF02	10 s	1900 × 1080 @24	low	Car fire
VF06	200 s	1280 × 720 @30	high	Car fire
Fi01	15 s	1280 × 720 @30	high	Fighting
Fi08	31 s	640 × 480 @30	high	Fighting
Sn06	23 s	1920 × 1080 @25	low	Snatching
Sn09	40 s	848 × 480 @30	high	Snatching

**Table 2 sensors-25-00251-t002:** Statistical overview of the 4 major categories of the UCF Crime dataset.

File Name	Video Length	Frames	Type of Anomaly
Road Accidens004_x264.mp4	11 s	347	Accident
Fighting007_x264.mp4	31 s	944	Fighting
Explosion046_x246.mp4	25 s	757	Explosion
Stealing020_x246.mp4	20 s	450	Stealing

**Table 3 sensors-25-00251-t003:** Model evaluation of 4 major categories (4MajCat) using the UCF dataset.

Model	Precision (%)	Recall (%)	F1-Score (%)	Accuracy (%)
DenseNet169	65.8	65.8	65.9	65.9
DenseNet201	68.1	68.2	68.2	68.2
DenseNet169 + Bi-LSTM	72.5	72.5	72.6	72.1
DenseNet201 + Bi-LSTM	78.8	78.5	78.7	78.5
Proposed CRBA Model	86.2	86.8	86.5	86.20

**Table 4 sensors-25-00251-t004:** Comparison of the proposed model with prior studies utilizing the UCF dataset.

Ref., Year	Accuracy (%)
[57], 2021	75.7
[58], 2022	74.53
[59], 2021	78.43
[60], 2023	81.01
[61], 2024	62.04
Proposed Model	86.20

**Table 5 sensors-25-00251-t005:** Model’s evaluation using RAD.

Model	Precision (%)	Recall (%)	F1-Score (%)	Accuracy (%)
DeneNet169	75.5	76.2	76.2	75.5
DenseNet201	78.2	77.8	78.5	78.6
DenseNet169 + Bi-LSTM	85.8	85.02	85.01	85.3
DenseNet201 + Bi-LSTM	88.2	88.3	88.6	88.5
Proposed CRBA Model	92.4	92.2	92.1	92.2

**Table 6 sensors-25-00251-t006:** The model’s accuracy across the UCF Crime dataset and RAD.

Model	UCF Crime	RAD
DenseNet169	65.9%	75.5%
DenseNet201	68.2%	78.6%
DenseNet169 + Bi-LSTM	72.1%	85.3%
DenseNet201 + Bi-LSTM	78.5%	88.5%

**Table 7 sensors-25-00251-t007:** The performance of the proposed CRBA model by class on the UCF Crime (4MajCat) and RAD.

Dataset	Class	Precision (%)	Recall (%)	F1-Score (%)	Accuracy (%)
UCF Crime	Accident	85.5	85.01	85.7	85.5
	Fight	86.7	86.8	86.4	86.5
	Explosion	87.1	87.5	87.2	87.1
	Stealing	86.0	86.5	86.2	86.2
	Average	86.2	86.8	86.5	86.2
RAD	Road Accident	92.01	91.75	91.8	92.2
	Car Fire	93.10	92.50	92.7	92.5
	Fighting	91.8	92.01	91.8	92.01
	Snatching	92.5	92.2	92.3	92.2
	Average	92.4	92.2	92.1	92.2

## Data Availability

The datasets are publicly available as the UCF dataset (https://www.kaggle.com/datasets/minhajuddinmeraj/anomalydetectiondatasetucf) and RAD (https://data.mendeley.com/datasets/8chk8vdn2z/1), accessed on 15 August 2024.

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
